# Photosynthetic Linear Electron Flow Drives CO_2_ Assimilation in Maize Leaves

**DOI:** 10.3390/ijms22094894

**Published:** 2021-05-05

**Authors:** Ginga Shimakawa, Chikahiro Miyake

**Affiliations:** 1Department of Biological and Environmental Science, Faculty of Agriculture, Graduate School of Agricultural Science, Kobe University, 1-1 Rokkodai, Nada, Kobe 657-8501, Japan; ginshimakawa@gmail.com; 2Core Research for Environmental Science and Technology, Japan Science and Technology Agency, 7 Goban, Chiyoda, Tokyo 102-0076, Japan

**Keywords:** photosynthesis, linear electron flow, C_4_ plants, P700 oxidation, ferredoxin

## Abstract

Photosynthetic organisms commonly develop the strategy to keep the reaction center chlorophyll of photosystem I, P700, oxidized for preventing the generation of reactive oxygen species in excess light conditions. In photosynthesis of C_4_ plants, CO_2_ concentration is kept at higher levels around ribulose 1,5-bisphosphate carboxylase/oxygenase (Rubisco) by the cooperation of the mesophyll and bundle sheath cells, which enables them to assimilate CO_2_ at higher rates to survive under drought stress. However, the regulatory mechanism of photosynthetic electron transport for P700 oxidation is still poorly understood in C_4_ plants. Here, we assessed gas exchange, chlorophyll fluorescence, electrochromic shift, and near infrared absorbance in intact leaves of maize (a NADP-malic enzyme C_4_ subtype species) in comparison with mustard, a C_3_ plant. Instead of the alternative electron sink due to photorespiration in the C_3_ plant, photosynthetic linear electron flow was strongly suppressed between photosystems I and II, dependent on the difference of proton concentration across the thylakoid membrane (ΔpH) in response to the suppression of CO_2_ assimilation in maize. Linear relationships among CO_2_ assimilation rate, linear electron flow, P700 oxidation, ΔpH, and the oxidation rate of ferredoxin suggested that the increase of ΔpH for P700 oxidation was caused by the regulation of proton conductance of chloroplast ATP synthase but not by promoting cyclic electron flow. At the scale of intact leaves, the ratio of PSI to PSII was estimated almost 1:1 in both C_3_ and C_4_ plants. Overall, the photosynthetic electron transport was regulated for P700 oxidation in maize through the same strategies as in C_3_ plants only except for the capacity of photorespiration despite the structural and metabolic differences in photosynthesis between C_3_ and C_4_ plants.

## 1. Introduction

In chloroplasts of plant leaves, photosynthetic CO_2_ assimilation is driven in the Calvin–Benson cycle utilizing NADPH and ATP produced by light energy [1]. In the photosynthetic electron transport system, light energy is absorbed by chlorophyll in photosystems (PS) I and II, producing the photo-oxidized reaction center chlorophylls, P700^+^ and P680^+^, to initiate photosynthetic linear electron flow (LEF) from PSII to PSI via the plastoquinone (PQ) pool, the cytochrome (Cyt) *b*_6_/*f* complex, and plastocyanin (PC). On the electron acceptor side of PSI, NADP^+^ is reduced to NADPH using electrons from PSI via ferredoxin (Fd) and Fd-NADP^+^ reductase. In PSII, H^+^ is released by H_2_O oxidation in the luminal side of the thylakoid membrane; in the Cyt *b*_6_/*f* complex, the Q-cycle pumps stromal H^+^ to the luminal side. Both generate a proton motive force (pmf) across the thylakoid membrane to produce ATP via the chloroplast ATP synthase [2]. In the Calvin–Benson cycle, ribulose 1,5-bisphosphate (RuBP) carboxylase/oxygenase (Rubisco) catalyzes the carboxylation of RuBP to produce two molecules of 3-phosphoglycerate (3-PGA), which are then metabolized in the Calvin–Benson cycle with NADPH and ATP. In this process, RuBP is regenerated. Meanwhile, Rubisco competitively catalyzes the oxygenation of RuBP, when available CO_2_ is limited in the presence of sufficient O_2_, to produce 3-PGA and 2-phosphoglycolate (2-PG). Finally, 2-PG is converted into 3-PGA in chloroplasts, peroxisomes, and mitochondria using Fd^−^ and ATP. These processes are the so-called photorespiration, in which CO_2_ is released from glycine in the mitochondria. The production and consumption of both NADPH and ATP are normally balanced to poise the redox state of the photosynthetic electron transport system in C_3_ plants [3,4]. In C_4_ plants, the capacity of photorespiration has been reduced during the evolutionary history [5,6] because the C_4_ plant leaves developed the CO_2_ concentrating mechanism, where CO_2_ is incorporated into phosphoenolpyruvate (PEP) in mesophyll cells and then transported in the form of malate into bundle sheath cells that specifically express Rubisco [7]. Malate is converted into CO_2_ and pyruvate around Rubisco by NADP-malic enzyme (ME) with NADP^+^ as the electron acceptor, and PEP is regenerated from pyruvate in mesophyll cells with two additional ATP [8,9]. Due to the structural and metabolic complexities, the regulation of photosynthetic electron transport has been poorly understood in C_4_ plants.

Maintaining P700 highly oxidized is a universal physiological response in photosynthetic organisms to prevent photo-oxidative damage of PSI derived from reactive oxygen species (ROS) by dissipating excess light energy as heat [10]. In contrast to PSII, which is also damaged by ROS [11,12], PSI is hardly impaired, but takes days or weeks to be recovered once it is damaged [13]. Therefore, it is essential for the protection of PSI to prevent the ROS generation within PSI by keeping P700 oxidized [10,14,15]. Here, we use the term “P700 oxidation” as the physiological response of photosynthetic organisms to keep the ratio of oxidized P700 to the total one higher, which can relieve the photo-excitation pressure in PSI because oxidized P700 immediately dissipates the excitation energy as heat. In intact plant leaves, P700 is commonly kept oxidized by a variety of molecular mechanisms in response to excess light conditions such as high light and CO_2_ limitation [14]. In C_3_ plants, P700 oxidation is supported by the suppression of electron transport in the Cyt *b*_6_/*f* complex, dependent on the difference of proton concentration across thylakoid membranes (ΔpH) [16,17,18]. Photorespiration replaces CO_2_ assimilation to function as an electron sink for P700 oxidation under limited CO_2_ conditions [19]. In C_4_ plants, there is little electron sink by photorespiration even at the CO_2_ compensation point, different from C_3_ plants [6,20]. It is still under debate how P700 remains oxidized in C_4_ plants when CO_2_ assimilation is suppressed. One important hypothesis that should be tested is that cyclic electron flow around PSI (CEF) functions to make ΔpH to keep P700 oxidized in C_4_ plants [21]. Since CEF is mediated by the electron transport from Fd^−^ to PQ, theoretically it is capable of pumping H^+^ from the stroma to the lumen of the thylakoid membrane in the Q-cycle; this also results in an additional ATP production that is not linked to NADP^+^ reduction [22]. In C_4_ plants, the ratio of PSI to PSII is much higher in isolated bundle sheath cells than in the mesophyll cells, which gives the hypothesis that CEF in bundle sheath cells contributes not only to keep P700 oxidized, but also to meet the additional ATP demand for C_4_ photosynthesis. Unfortunately, there is currently no method available to directly measure the electron transport rate via CEF. The CEF activity has been indirectly estimated mainly from the quantum yield of PSI, which is inevitably under/overestimated dependent on the redox state of PC [3]. Therefore, the extent of CEF activity remains controversial.

In this study, we evaluated gas exchange, chlorophyll fluorescence, electrochromic shift (ECS), and near infrared (NIR) absorbance in the NADP-ME subtype of C_4_ plants maize to investigate how photosynthetic electron transport is regulated for P700 oxidation. Although maize does not have significant capacity of photorespiration, the linear relationships among CO_2_ assimilation rate, LEF, P700 oxidation, ΔpH, and the Fd^−^ oxidation rate were observed as shown in the C_3_ plant field mustard (Komatsuna, hereafter mustard). The ratio of PSI to PSII was estimated in vivo approximately 1:1 in the both C_3_ and C_4_ plants. These results suggest that maize keeps P700 oxidized tightly associated with LEF same as C_3_ plants, but strongly relies on the ΔpH-dependent suppression of electron transport instead of photorespiratory electron sink.

**Figure 1 ijms-22-04894-f001:**
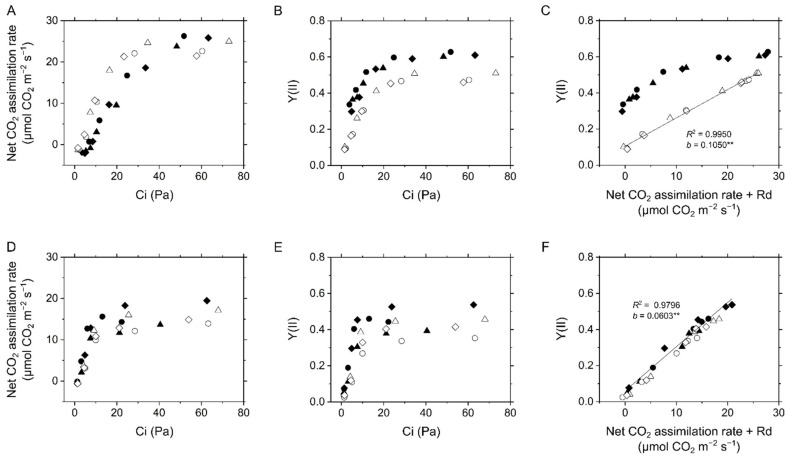
Photosynthetic CO_2_ assimilation and linear electron flow in the C_3_ plant mustard (**A**–**C**) and the C_4_ plant maize (**D**–**F**). (**A**,**D**) Net CO_2_ assimilation rate at various intercellular CO_2_ partial pressures (Ci). (**B**,**E**) Effective quantum yield of PSII, Y(II), at various Ci. (**C**,**F**) Relationship of Y(II) with CO_2_ assimilation rate. Dark respiration rate is presented as Rd. Experiments were conducted independently three times as shown in different symbols (biological replicates) at 21 kPa (closed symbols) and 1 kPa O_2_ (open symbols). Solid lines represent estimated linear regression of the data at 1 kPa (**C**) and 21 kPa O_2_ (F) (*R*^2^, coefficient of determination). The *y*-intercepts (*b*) were tested based on the null hypothesis: ** *p* < 0.005.

## 2. Results

Photosynthetic CO_2_ assimilation and dark respiration were analyzed in both the C_3_ plant mustard and the C_4_ plant maize simultaneously with chlorophyll fluorescence, ECS, and NIR absorbance. In this study, we analyzed in vivo photosynthetic parameters at a constant light intensity and different CO_2_ partial pressures to simply investigate the effects of limitation of electron sink on the photosynthetic electron transport. Additionally, the C_3_ and C_4_ intact leaves were measured at atmospheric (21 kPa) and low (1 kPa) O_2_, where photorespiration is suppressed. We note that in maize CO_2_ is first incorporated into PEP in mesophyll cells, different from C_3_ plants, but the decrease of the CO_2_ partial pressure in the intercellular space (Ci) finally results in the limitation of the carboxylation reaction of Rubisco in bundle sheath cells.

In mustard, the net CO_2_ assimilation rate was higher at 1 kPa than at 21 kPa O_2_ when available CO_2_ was limited (Figure 1A), whereas the effective quantum yield of PSII, Y(II), was kept high uncoupled from CO_2_ assimilation at 21 kPa O_2_ (Figure 1B,C). However, maize did not show any difference in net CO_2_ assimilation and Y(II) between different O_2_ partial pressures (Figure 1D,E), which is in accordance with a small contribution, if any, of photorespiration to the capacity of electron sink. As a result, Y(II) has a linear relationship with the sum of net CO_2_ assimilation rate and dark respiration rate (Rd) in maize (Figure 1F). These typical photosynthetic characteristics of C_3_ and C_4_ plants have already been established in previous studies, and extra Y(II) to net CO_2_ assimilation rate in mustard is known to be due almost exclusively to photorespiration [6,20,23,24,25,26]. Overall, Y(II) reflects LEF mainly derived from CO_2_ assimilation and photorespiration in angiosperms [17,27,28]. The inferred reduction level of the PQ pool (1-qL) and non-photochemical quenching (NPQ) were also calculated from chlorophyll fluorescence in mustard and maize, both of which increased at low Ci (Appendix A). In mustard, 1-qL at 1 kPa O_2_ increased further at low Ci, whereas NPQ slightly decreased in that condition (Appendix A).

**Figure 2 ijms-22-04894-f002:**
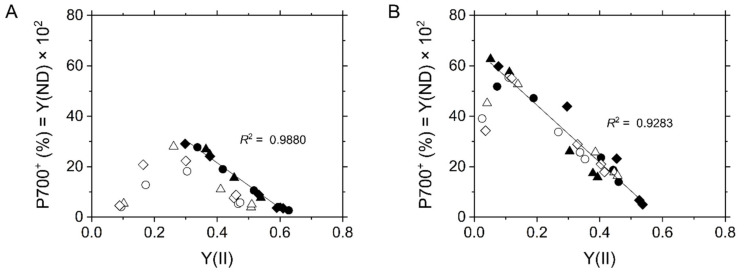
Relationship of P700 oxidation with effective quantum yield of PSII, Y(II), at various intercellular CO_2_ partial pressures in the C_3_ plant mustard (**A**) and the C_4_ plant maize (**B**). Experiments were conducted independently three times as shown in different symbols (biological replicates) at 21 kPa (closed symbols) and 1 kPa O_2_ (open symbols). Solid lines represent estimated linear regression of the data at 21 kPa O_2_ (*R*^2^, coefficient of determination).

**Figure 3 ijms-22-04894-f003:**
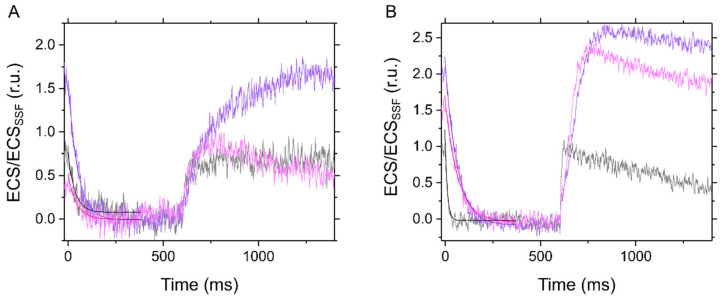
Dark-interval relaxation kinetics of electrochromic shift (ECS) in the C_3_ plant mustard (**A**) and the C_4_ plant maize (**B**) under ambient air (40 Pa CO_2_, 21 kPa O_2_; black), low CO_2_ (1 Pa CO_2_, 21 kPa O_2_; purple), and low CO_2_/O_2_ (1 Pa CO_2_, 1 kPa O_2_; pink). Red actinic light (550 μmol photons m^−2^ s^−1^) was turned off at 0 ms for 600 ms during the steady-state photosynthesis. The ECS values were normalized by the ECS amplitude induced by 5 µs short saturation flash (ECS_SSF_). The kinetics were fit to mono exponential decay (*R*^2^, coefficient of determination: 0.7017, 0.9454, and 0.6868 in A; 0.8814, 9868, and 0.9801 in B, respectively).

Next, we evaluated the oxidation of P700 in the relationship with LEF reflected in Y(II) (Figure 2). In mustard at 21 kPa O_2_, the oxidation level of P700 increased as there was a decrease in Y(II) (Figure 2A) although the PQ pool was likely to be reduced at low Ci (Appendix A). That is, photosynthetic electron transport is suppressed between PSII and PSI, presumably at the Cyt *b*_6_/*f* complex. However, at 1 kPa O_2_, P700 started to be kept reduced when Y(II) <0.3 (Figure 2A), which suggests that the electron sink by photorespiration is required for P700 oxidation [19]. Unlike mustard, P700 continued to remain oxidized in maize even at 1 kPa O_2_ except under an extreme CO_2_ limitation (<1.5 Pa) (Figure 2B). The inverse proportional relationship in the redox states of the PQ pool and P700 suggested that LEF was limited by the suppression of electron transport at the Cyt *b*_6_/*f* complex in maize as in C_3_ plants (Figure 2 and Appendix A).

**Figure 4 ijms-22-04894-f004:**
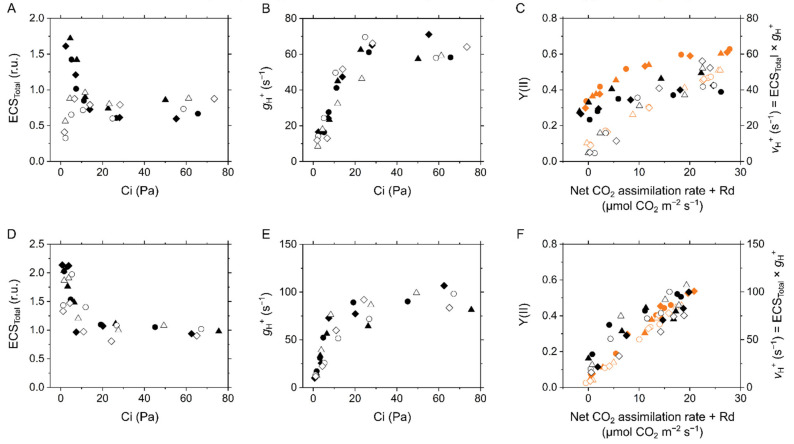
Electrochromic shift (ECS) parameters in the C_3_ plant mustard (**A**–**C**) and the C_4_ plant maize (**D**–**F**). (**A**,**D**) Total ECS values reflecting proton motive force at various intercellular CO_2_ partial pressures (Ci). (**B**,**E**) Proton conductance of the chloroplast ATP synthase (*g*_H_^+^) at various Ci. (**C**,**F**) Relationship of proton efflux rate via the ATP synthase (*v*_H_^+^) with CO_2_ assimilation rate. Effective quantum yield of PSII, Y(II), are also shown in orange symbols. Dark respiration rate is presented as Rd. Experiments were conducted independently three times, as shown in different symbols (biological replicates) at 21 kPa (closed symbols) and 1 kPa O_2_ (open symbols).

To investigate molecular mechanisms for P700 oxidation in response to the limitation of electron sink, we evaluated the thylakoid membrane potential by ECS analysis in the transition from light to dark at the steady-state photosynthesis (Figure 3). We note that the ECS parameters are dependent on the properties of the leaves, not only the density of chloroplasts, but also the content of light-harvesting complexes that house the shifted pigments and are normally associated with PSII. Therefore, it is difficult to make any quantitative conclusions for the differences in the amplitudes of ECS parameters between mustard and maize. In this study, the ECS values are normalized by the ECS amplitude induced by the short-saturation flash (ECS_SSF_; see the Section 4). In the so-called dark-interval relaxation kinetics, pmf in the light is probed by the total rapid (<1 s) change in the ECS signal upon rapidly switching off the light (ECS_Total_), which increased with the limitation of CO_2_ in both mustard and maize at 21 kPa O_2_ (Figure 4A,D). However, in mustard, ECS_Total_ was not enhanced in response to the suppression of CO_2_ assimilation at 1 kPa O_2_, where photorespiration is inhibited (Figure 4A). That is, photorespiration functions as the electron sink to sustain LEF, supporting the increase of pmf in C_3_ plants [19]. Nevertheless, we found the increase of ECS_Total_ regardless of photorespiratory electron sink in maize (Figure 4D). Proton conductance of the ATP synthase (*g*_H_^+^) is calculated as the rate constant of the mono-exponential ECS decay [29], which increased with Ci and was then saturated as in the trend of CO_2_ assimilation rate regardless of O_2_ partial pressures in both mustard and maize (Figure 4B,E). Further, the initial decay rate of the ECS changes is termed as relative light-driven proton flux through the chloroplast ATP synthase, the so-called *v*_H_^+^ [30], which seems to be correlated with the sum of net CO_2_ assimilation rate and Rd at 1 kPa O_2_ in mustard (Figure 4C). In mustard at 21 kPa O_2_, a part of *v*_H_^+^ was uncoupled from the CO_2_ assimilation rate (Figure 4C), like the case of Y(II) (Figure 1C), which suggested that *v*_H_^+^ has almost the linear relationship with LEF derived from CO_2_ assimilation and photorespiration in C_3_ intact leaves. In the case of maize leaves, the relationship of *v*_H_^+^ with photosynthetic CO_2_ assimilation may also be linear at both 21 and 2 kPa O_2_ (Figure 4F). However, we note the possibility that *v*_H_^+^ decreased with photosynthetic CO_2_ assimilation larger in lower Ci conditions (Figure 4F), implying that ATP is utilized in a different manner at different Ci levels.

**Figure 5 ijms-22-04894-f005:**
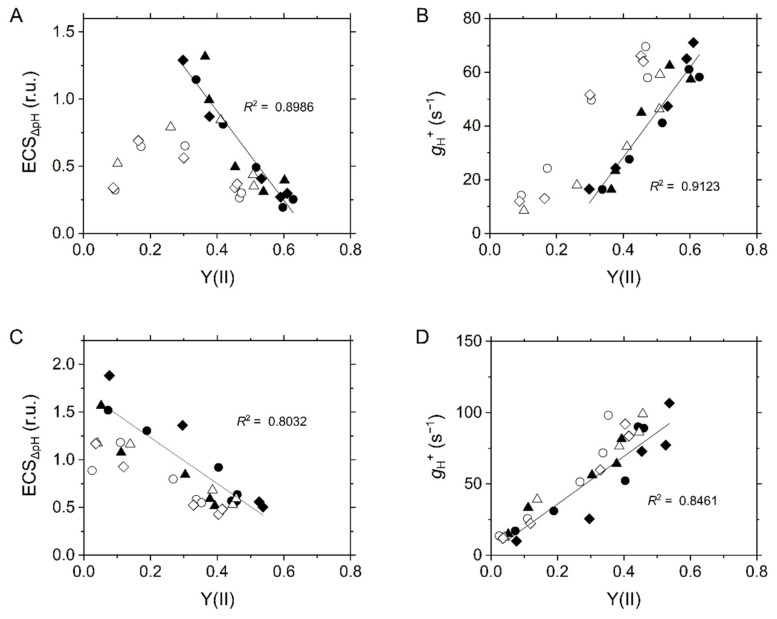
Relationships of the ECS values reflecting the difference of proton concentration across the thylakoid membrane (ECS_ΔpH_); (**A**,**C**) and the proton conductance of the chloroplast ATP synthase (*g*_H_^+^; B, D) with effective quantum yield of PSII, Y(II), at various intercellular CO_2_ partial pressures in the C_3_ plant mustard (**A**,**B**) and the C_4_ plant maize (**C**,**D**). We note that ΔpH and *g*_H_^+^ were separately measured from Y(II) at the same ambient CO_2_ partial pressures. Experiments were conducted independently three times as shown in different symbols (biological replicates) at 21 kPa (closed symbols) and 1 kPa O_2_ (open symbols). Solid lines represent estimated linear regression of the data at 21 kPa O_2_ (*R*^2^, coefficient of determination).

By definition, the ECS_Total_ includes two components: ∆pH and transmembrane difference in the electric potential (∆Ψ), which can be distinguished by the post-illumination transient change in ECS, and the former component triggers the suppression of electron transport in the Cyt *b*_6_/*f* complex [31]. Like P700 oxidation, ECS_∆pH_ showed the linear relationship with LEF at 21 kPa O_2_ in both mustard and maize. The formation of ∆pH fraction of pmf was suppressed at 1 kPa O_2_ in mustard at low Ci and in maize at very low Ci (Figure 5A,C). The increase of ECS_∆pH_ was associated with NPQ (Appendix A), which was in agreement with that NPQ at PSII is stimulated by ∆pH [18,29,32]. There were two possibilities for the lumen acidification: (1) H^+^-pumping from stroma into the thylakoid lumen was promoted; or (2) H^+^-leakage from the lumen to stroma was blocked. Both mustard and maize showed the linear relationship of *g*_H_^+^ with LEF reflected in Y(II) (Figure 5B,D), which is consistent with previous reports [17,18,19,33] and suggests that limiting *g*_H_^+^ leads to the increase in ∆pH resulting in P700 oxidation.

**Figure 6 ijms-22-04894-f006:**
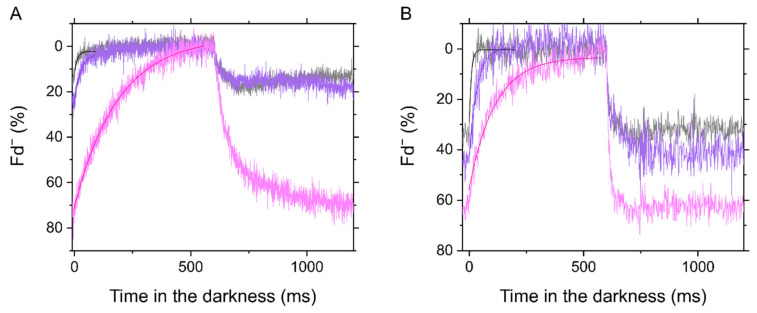
Dark-interval relaxation kinetics of ferredoxin (Fd^−^) in the C_3_ plant mustard (**A**) and the C_4_ plant maize (**B**) under ambient air (40 Pa CO_2_, 21 kPa O_2_; black), low CO_2_ (1 Pa CO_2_, 21 kPa O_2_; purple), and low CO_2_/O_2_ (1 Pa CO_2_, 1 kPa O_2_; pink). Red actinic light (550 μmol photons m^−2^ s^−1^) was turned off at 0 ms for 600 ms during the steady-state photosynthesis. The kinetics were fit to mono exponential decay (*R*^2^, coefficient of determination: 0.6628, 0.8816, and 0.9788 in A; 0.6612, 0.8307, and 0.9339 in B).

In this study, we investigated the redox state of Fd at the steady state of photosynthesis in both mustard and maize using a Klas-NIR spectrophotometer. The maximum amplitude of photo-reducible Fd was determined by the standard method in advance as shown in Appendix A. The decay of Fd^−^ in the transition from light to dark was mono-exponentially fit, giving the amplitude and the oxidation rate of Fd^−^ (Figure 6). Exceptionally, a biphasic decay might be recognized in the kinetics at 1 Pa CO_2_ and 1 kPa O_2_ in maize (Figure 6B). The slow component remains to be identified, but it was clearly negligible compared to LEF. In mustard, Fd was strongly reduced under CO_2_ limitation at 1 kPa O_2_, whereas it was totally kept oxidized at 21 kPa O_2_ (Figure 6A and Figure 7A), indicating that an O_2_-dependent alternative electron sink relieves the acceptor-side limitation of PSI. Photorespiration and Mehler reaction are considered as the molecular mechanism related to the electron sink. Since the electron flux capacity via photorespiration is significantly larger than that via Mehler reaction at a CO_2_ compensation point in C_3_ plants [20,28], the oxidation of Fd seems to be mainly due to photorespiration. In maize, Fd^−^ was gradually accumulated with the decrease in Ci at higher levels at 1 kPa O_2_ than at 21 kPa O_2_ (Figure 6B and Figure 7C). The intact leaves of mustard showed a linear relationship of Fd^−^ oxidation rate with Y(II), and its *y*-intercept was close to zero (Figure 7C), which is in agreement with the recent study on C_3_ plants [34]. Further, the same trend was observed also in maize leaves (Figure 7D). That is, LEF estimated from Y(II) clearly corresponds to the electron transport via Fd in the situations where ∆pH increased and P700 was oxidized in response to the limitation of electron sink in both mustard and maize.

We also plotted effective quantum yield of PSI, Y(I), against Y(II), which is a conventional method to evaluate CEF activity used in numerous previous and recent reports. In mustard at 21 kPa O_2_, we observed a linear relationship between Y(I) and Y(II) with extra Y(I) to Y(II) at lower Ci (Figure 8A). Additionally, Y(I) showed no linear relationship with Y(II) in mustard at 1 kPa O_2_ and maize at both 21 and 1 kPa O_2_, resulting in the extra Y(I) (Figure 8A,C), different from the relationship of Fd^−^ oxidation rate with Y(II). Here, we also evaluated the oxidation of PC and plotted it against Y(II). In mustard, PC was kept more oxidized at lower Ci, but it was reduced where photorespiration was inhibited (Figure 8B), like P700 (Figure 2A). In maize, the redox state of PC gave a curved plot, indicating the relationship between PC oxidation and Y(II) changed on around 0.2 of Y(II) (Figure 8D). Interestingly, the extra Y(I) to Y(II) was coincided with the oxidation of PC (Figure 8, Appendix A).

The linear relationships among CO_2_ assimilation rate, Y(II), P700 oxidation, ∆pH, and Fd^−^ oxidation rate (Figure 1, Figure 2, Figure 5 and Figure 7) suggested that the regulation of photosynthetic electron transport is tightly associated with LEF in C_4_ plants, similarly to C_3_ plants, which is somehow unexpected if maize has CEF in bundle sheath chloroplasts. Originally, CEF in C_4_ plants is considered based on the biochemical fact that there is less grana structure containing PSII in the isolated bundle sheath cells. It should be noted that in this study we analyzed a variety of photosynthetic parameters at the scale of intact leaves that is the mixture of mesophyll and bundle sheath cells. Here, we spectroscopically estimated the ratio of PSI to PSII in vivo in the intact leaves of mustard and maize. A short-saturation flash induces ECS dependent on the photochemical reaction at PSII and PSI. During far-red light illumination, where PSI is selectively excited, the short-saturation flash induces ECS presumably originated only from PSII [35]. The ECS amplitude, i.e., ECS_SSF_, decreased with the intensity of far-red light, finally reaching approximately 50% of the initial amplitude in both mustard and maize (Figure 9), indicating that the ratio of PSII to PSI is about 1:1 at the scale of leaves. The same results were obtained from field-grown sunflower (C_3_ plant) and maize (Appendix A).

To further test if there is significant amount of PSI uncoupled from PSII in bundle sheath cells of maize, the electron transport from PSII to PSI was roughly estimated in vivo using a Klas-NIR spectrophotometer. A short-saturation flash was applied to excite PSII and PSI after accumulating P700^+^ by the far-red light illumination, resulting in the decay kinetics of P700^+^ (Figure 10). The P700^+^ reduction kinetics were likely to be involved in more than two components, but the rapid decay within 100 ms should reflect the electron transport from PSII [36]. Interestingly, a part of P700^+^ (10−20%) took more than second to be reduced, which may be due to a redox equilibration between PC and P700 as implied from the reduction of a part of PC (ca. 20%) within 100 ms (Figure 10A,D). The redox state of Fd did not change in response to the short-saturation flash (Figure 10A,D). The ratio of P700^+^ rapidly reduced was rather larger in maize (ca. 80%) than in mustard (ca. 65%), both of which increased with the length of the short-saturation flash (5−50 μs) to approximately 90% (Figure 10B,C,E,F). These results implied that there is no significant amount of PSI uncoupled from PSII in mustard and maize at the scale of the intact leaves. It should be noted that P700 was kept more reduced in maize than in mustard during far-red light illumination (Figure 10A,D, and Appendix A), and a part of P700^+^ (ca. 20%) was immediately reduced within 50 μs just after the illumination with 5 μs short-saturation flash (Figure 10D), which was likely to be too fast for CEF, considering the turnover of the Cyt *b*_6_/*f* complex [37].

**Figure 9 ijms-22-04894-f009:**
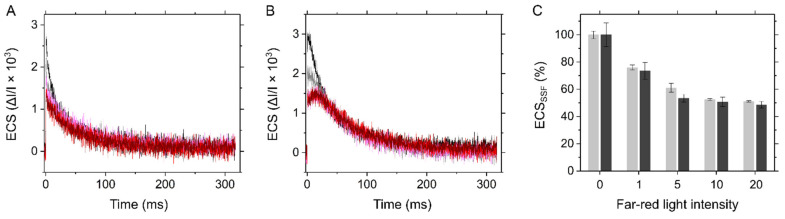
Electrochromic shift induced by a 5 μs-short saturation flash (ECS_SSF_) during far-red light illumination in the C_3_ plant mustard (**A**) and the C_4_ plant maize (**B**). Far-red light was provided at various intensities (0, black; 1, grey; 5, pink; 10, red; and the maximum 20, wine red; the values defined by the Walz software). (**C**) The flash-induced ECS changes normalized by the values without far-red light illumination as 100%. The data of mustard (light grey) and maize (dark grey) are shown as the mean with the standard deviation (*n* = 3, biological replicates).

## 3. Discussion

We characterized the in vivo regulatory mechanisms of photosynthetic electron transport for P700 oxidation in maize. Whereas P700 oxidation in maize showed the different O_2_ dependency from that in mustard, the regulatory process could be explained same as C_3_ plants without considering the metabolic complexity in co-operation of mesophyll and bundle sheath cells in C_4_ plants. Oxidation of P700 is the universal strategy for photosynthetic organisms to suppress the generation of ROS at the acceptor side of PSI, which is mainly regulated in C_3_ plants by the donor-side mechanism, i.e., the ΔpH-dependent suppression of electron transport at the Cyt *b*_6_/*f* complex, and by the alternative electron sink, photorespiration. Since maize did not show photorespiration as an alternative electron sink, LEF was linearly suppressed with the decrease in photosynthetic CO_2_ assimilation (Figure 1). Nevertheless, P700 is kept oxidized with the suppression of CO_2_ assimilation, which is linearly associated with the increase in ECS_ΔpH_ by limiting *g*_H_^+^ (Figure 5). In these processes, the Fd^−^ oxidation rate showed the linear relationship with Y(II) (Figure 7). The metabolic compartmentation in C_4_ photosynthesis and the higher ratio of PSI to PSII in isolated bundle sheath cells have made it complicated to consider the regulation of photosynthetic electron transport in C_4_ plants. However, the ratio of PSI to PSII was estimated almost 1:1 at the scale of the intact maize leaves, similarly to C_3_ plants (Figure 9). Further, the ratio of P700^+^ rapidly reduced by a short-saturation flash was similar between mustard and maize (Figure 10). All these results supported that the robustness of P700 oxidation is tightly associated with LEF in the C_4_ plant maize, as previously proposed in C_3_ plants [17].

**Figure 10 ijms-22-04894-f010:**
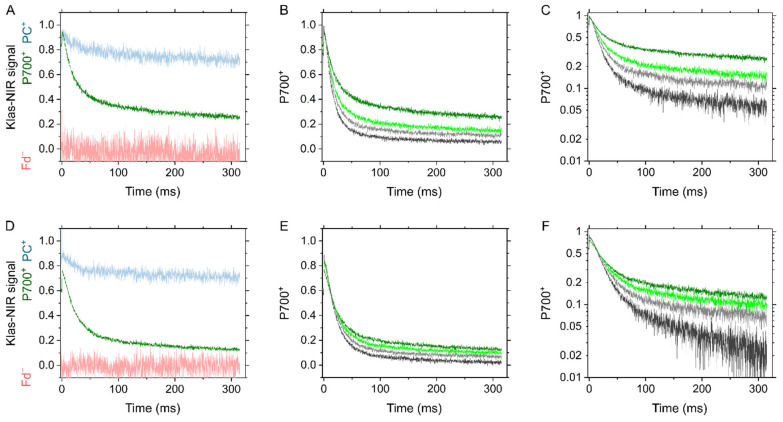
Effects of a short-saturation flash on the redox state around PSI in the C_3_ plant mustard (**A**–**C**) and the C_4_ plant maize (**D**–**F**). (**A**,**D**) Reduction kinetics of P700^+^ (green) in response to a 5 μs-short saturation flash after far-red light illumination for 10 s. Far-red light was provided at the maximum intensity (20, the value defined by the Walz software). Deconvoluted signals to plastocyanin (PC^+^, blue) and ferredoxin (Fd^−^, red) are also shown. All the Klas-NIR signals were normalized within the range from the minimum 0 to the maximum 1. The maximum reduction/oxidation levels of each component were determined as shown in Appendix A. The signal to Fd^−^ has negative values. (**B**,**E**) Reduction kinetics of P700^+^ by a short saturation flash at different lengths (5, green; 10, light green; 20, light grey; and 50 μs, grey, respectively) after far-red light illumination for 10 s. The kinetics are also shown with the logarithmic scale at *y*-axis (C, F). The representative traces of independent experiments (*n* = 3, biological replicates) are shown.

Instead of the photorespiratory electron sink, maize strongly relies P700 oxidation on the ΔpH-dependent suppression of electron transport at the Cyt *b*_6_/*f* complex (Figure 2B and Figure 5C). The decrease in *g*_H_^+^ was linearly correlated with LEF reflected in Y(II) (Figure 5D) and Fd^−^ oxidation rate (Figure 7D). These results suggest that lumen acidification for P700 oxidation was attributed to the decrease of *g*_H_^+^, but not to an additional H^+^-pumping from stroma to the thylakoid lumen, for example, by CEF (also see the discussion in the next section). This conclusion is consistent with the report by Kiirats et al. [33] showing the linear relationship between photosynthetic O_2_ evolution and *g*_H_^+^ in all three C_4_ subtypes. The different dependencies of P700 oxidation on the regulation of *g*_H_^+^ led to the different threshold of LEF for keeping P700 oxidized between mustard and maize. Contrary to mustard, which needed about 50% of the maximum Y(II) for P700 oxidation, maize did not exhibit breakdown of P700 oxidation even when Y(II) was close to zero (Figure 2). The linear proportional relationship between Y(II) and P700 oxidation indicated that the C_4_ plant maize does not require an electron sink for P700 oxidation (Figure 2B). The insensitivity to O_2_ of P700 oxidation has been observed also at various irradiances at CO_2_-saturated conditions [38]. Nevertheless, an extreme condition of 1 kPa O_2_ and very low CO_2_ (<1.5 Pa) partially disturbed P700 oxidation in maize (Figure 2B), which might be due to the slight but certain electron flux via photorespiration or the Mehler reaction in C_4_ plants [24,39]. Overall, C_4_ plants ultimately utilize O_2_ for P700 oxidation maybe in some extreme conditions. In C_3_ plants, photorespiration relieves the electron transport limitation on the acceptor side of PSI and also keeps producing ΔpH (Figure 2A and Figure 5A,C, and Appendix A). It should be trade-off to rely P700 oxidation mainly on the regulations on the donor or acceptor sides of PSI. The contribution of the regulation of *g*_H_^+^ to P700 oxidation would be different among different types of C_4_ plants and C_3_–C_4_ intermediates associated with the capacity of photorespiration [6]. Whereas P700 oxidation with the suppression of photosynthesis is the phenomenon commonly observed in a variety of photosynthetic organisms, the dominant molecular mechanisms are rich in diversity, which has been already diversified among different species of cyanobacteria, the progenitor of oxygenic photosynthesis [15], and has changed during the evolutionary history of photosynthetic green and red plastid lineages. Interestingly, the strategy for P700 oxidation in maize can be categorized on the view of O_2_-usage into the same type of that in some secondary algae derived from red algae, which do not need O_2_ for P700 oxidation [40].

**Figure 11 ijms-22-04894-f011:**
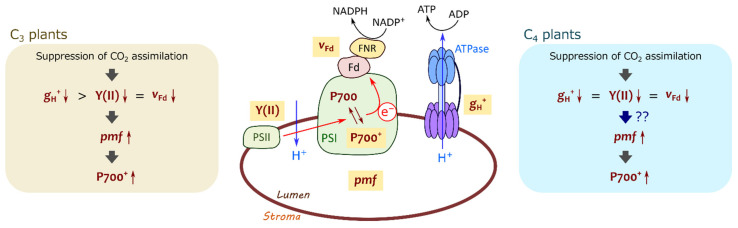
A brief illustration of mechanism for P700 oxidation in C_3_ and C_4_ plants. In C_3_ plants, proton conductance of chloroplast ATP synthase (*g*_H_^+^) decreases with the suppression of photosynthetic CO_2_ assimilation greater than photosynthetic linear electron flow reflected in effective quantum yield of PSII, Y(II), and ferredoxin (Fd^−^) oxidation rate (*v*_Fd_), resulting in the increase in proton motive force (pmf) to induce P700 oxidation. In C_4_ plants, pmf increases with the suppression of photosynthetic CO_2_ assimilation, although *g*_H_^+^ decreases concomitantly with Y(II) and *v*_Fd_.

What is the determinant for the strong contribution of the ΔpH-dependent donor side mechanism to P700 oxidation in maize? Lumen acidification should be controlled by both H^+^-pumping and H^+^-consumption rates in plant leaves. Based on the results suggesting that the CEF activity via Fd was negligible (Figure 7B,D), the H^+^-pumping rate into the thylakoid lumen is estimated from LEF with the rate constant (k_H_^+^), which is correlated with Y(II). Meanwhile, the H^+^-consumption rate is equal to the ECS parameter *v*_H_^+^, being likely to reflect ATP consumption by CO_2_ assimilation and photorespiration in C_3_ plants. Finally, the change in ECS_Total_ is presented as the following Equation (1) in C_3_ plants [17]:d(ECS_Total_)/dt = (k_H_^+^ × LEF)−(ECS_Total_ × *g*_H_^+^)(1)

At the steady state: ECS_Total_ = (k_H_^+^ × LEF)/*g*_H_^+^.

That is, lumen acidification occurs where LEF is sustained more than *g*_H_^+^. In mustard, the decrease in Y(II) (ca. 50%) was much smaller than that in *g*_H_^+^ (ca. 85%) at 21, but not at 1 kPa O_2_ (Figure 5B), indicating that photorespiration contributes to sustaining LEF, resulting in the H^+^ accumulation into the thylakoid lumen (Figure 5A and Figure 11) [19,41]. In maize, LEF decreased almost concomitantly with *g*_H_^+^, different from mustard (Figure 5D) [33]. Nevertheless, H^+^ was accumulated in the thylakoid lumen to cause P700 oxidation at both 21 and 1 kPa O_2_ (Figure 2B and Figure 5C). Overall, unlike C_3_ plants, ECS_Total_ in maize cannot be formulated as the above equation (Figure 11). Possibly, there is an additional mechanism to pump H^+^ into the thylakoid lumen independent of the electron transport via Fd, which remains to be further investigated in future works. It should be also noted that the relationship of *v*_H_^+^ with photosynthetic CO_2_ assimilation was different from that of Y(II) (Figure 4F), which is presumably due to the different ATP consumption for regenerating PEP at various Ci. The strong contribution of the ΔpH-dependent donor side mechanism to P700 oxidation in maize should be also related to the different ATP utilization between C_3_ and C_4_ plants.

The present results do not support the hypothesis that CEF is driven in the bundle sheath cells of NADP-ME subtype of C_4_ plants at a comparable flux to LEF in the mesophyll cells. Although there is the report that the PSII activities in the bundle sheath cells of NADP-ME subtype of C_4_ plants, including maize, sorghum, and Flaveria, are almost equal to those in the mesophyll cells [42], numerous studies have followed and documented that PSII activity is nearly negligible in isolated bundle sheath cells in NADP-ME subtype of C_4_ plants, especially in maize [43,44,45,46,47,48,49,50], having constructed the dogma that CEF is responsible for the additional ATP production in the bundle sheath cells of C_4_ plants. Actually, recent modelling studies select the CEF model for simulating C_4_ photosynthesis [51]. However, it should be noted that these studies follow the ratio of PSI to PSII in the experimentally differentiated mesophyll and bundle sheath cells, except for the semi-quantitative evaluation by immunocytology [52]. Overall, the PSI:PSII ratio has yet not been quantitatively understood at the scale of intact leaves, and there is still no evidence for the energetic contribution of CEF to photosynthesis in C_4_ plants. Another problem is that in vivo CEF activity has been generally evaluated using the effective quantum yield of PSI that is easily under/overestimated (Figure 8) [3]. Considering the experimental results by the previous and present studies, we propose that the amount of PSI in bundle sheath cells is presumably much smaller than that in mesophyll cells at the scale of C_4_ intact leaves.

It is still unclear how the additional ATP demand is met in C_4_ photosynthesis. In one of the important landmarks in C_4_ photosynthesis research, Chapman et al. (1980) have shown that in the presence of exogenously added malate photosynthetic CO_2_ assimilation proceeds in the isolated bundle sheath cells of maize with PSII inhibited by 3-(3,4-dichlorophenyl)-1,1-dimethylurea (DCMU). The PSII-independent ATP production can reach about 80% of the ATP production of the illuminated cells in the in vitro system [45], which clearly suggests that there is a malate-dependent ATP source in the cells. It should be also considered that the NAD(P)H dehydrogenase complex plays an important role for C_4_ photosynthesis [53]. It has been recently accepted that the intermediates of the Calvin–Benson cycle shuttle between the mesophyll and bundle sheath cells with certainly large fluxes [50,54], which can make it more complicated how much ATP and NADPH are needed, respectively, in these two types of cells in the light. In both mustard and maize, ECS_Total_ was formed to some extent even in the conditions where photosynthesis and photorespiration are almost completely suppressed under 1 kPa O_2_ and very low Ci (Figure 4A,D), which suggested that ATP is, at least, not limited. We also note that the stoichiometry of ATP and NADPH may not need to be necessarily satisfied because excess NADPH should not be accumulated as long as P700 oxidation system works [3].

For the last decades, CEF has been frequently evaluated by the comparison between Y(II) and Y(I). The extra Y(I) to Y(II) has been believed to be a conventional indicator to CEF activity. Nevertheless, recently it has been clearly shown that Y(I) can be easily over-estimated by the oxidation of PC [3]. In this study, the extra Y(I) was totally coincided with the PC oxidation in both mustard and maize at various CO_2_ and O_2_ partial pressures (Figure 8). These facts indicate that Y(I) must not be utilized to evaluate CEF activity. A variety of alternative methods, including the comparison of Fd^−^ oxidation rate with Y(II), should be considered in future works.

## 4. Materials and Methods

### 4.1. Plant Materials

Field mustard (Komatsuna, *Brassica rapa*) and maize (*Zea mays*) were grown under long-day conditions (14 h-light, 24 °C, 300 µmol photons m^−2^ s^−1^, white fluorescent lamp/10 h-dark, 22 °C). Seeds were planted in pots that contained a 5:3:2 mix of Metro-Mix 350 (Sun Gro Horticulture, Agawam, MA, USA), Akadama, and vermiculit with 1000-fold diluted Hyponex solution (Hyponex, Osaka, Japan) used as a watering solution. For the experiments in Appendix A, the field-grown sunflower (*Helianthus annuus*) and maize were used.

### 4.2. Gas Exchange, Chlorophyll Fluorescence, and Spectroscopic Analyses

Exchanges of CO_2_ and H_2_O were measured using a GFS-3000 equipped with a 3010-DUAL gas exchange chamber (Walz, Effeltrich, Germany) in which ambient air was saturated with water vapor at 18.0 ± 0.1 °C, and the leaf temperature was maintained at 25 ± 2 °C. Ci was calculated based on the previous report [55].

Chlorophyll fluorescence and near infrared absorbance were simultaneously measured coupled with gas exchange analysis using a Klas-NIR spectrophotometer (Walz) [56]. Chlorophyll fluorescence parameters were calculated as follows [57]: F_o_, minimum fluorescence from a dark-adapted leaf; F_m_′, maximum fluorescence from a light-adapted leaf; F′, fluorescence emission from a light-adapted leaf; Y(II) = (F_m_′ − F′)/F_m_′, effective quantum yield of PSII; qL = (F_m_′ − F′)/(F_m_′ − F_o_′) × (F_o_′/F′), fraction of “open” PSII centers (with Q_A_ oxidized) on the basis of a lake model for the PSII photosynthetic apparatus; NPQ = (F_m_ − F_m_′)/F_m_′, non-photochemical quenching. Pulse-amplitude modulated green measuring light (540 nm, <0.1 µmol photons m^−2^ s^−1^) was used. To obtain F_m_′, a saturation flash (630 nm, 8000 µmol photons m^−2^ s^−1^, 300 ms) was applied. Red actinic light (630 nm, 550 µmol photons m^−2^ s^−1^) was supplied using a chip-on-board LED array. The signals for P700^+^, PC^+^, and Fd^−^ were calculated based on the deconvolution of four pulse-modulated dual-wavelength difference signals in the near infrared region (780–820, 820–870, 840–965, and 870–965 nm). The redox state of P700 was evaluated as the ratio of P700^+^ to the total P700, termed Y(ND) [58]. Both P700 and PC are kept completely reduced, and Fd is fully oxidized in dark conditions. For the determination of total photo-oxidizable P700 and PC, the saturation flash was applied after 10 s illumination with a far-red light (740 nm; Appendix A) [59]. Total photo-reducible Fd was determined by the illumination with a red actinic light (450 µmol photons m^−2^ s^−1^) after plant leaves were adapted to the dark for 5 min (Appendix A) [59]. For the analysis of a dark-interval relaxation kinetics, the red actinic light (550 µmol photons m^−2^ s^−1^) was temporarily turned off for 600 ms at the steady-state photosynthesis [34]. The oxidation rate of Fd^−^ was estimated as relative values by a Klas-NIR spectrophotometer by a linear fitting for the initial decay of Fd^−^. Recently, it has been shown that the reduced iron-sulfur clusters in PSI, F_A_/F_B_, can contribute to the Klas-NIR signal attributed to Fd^−^ in almost the same manner in vitro [60]. In this study, we concluded that the F_A_/F_B_ contribution to the Fd signal was not critical problem in the measurement for the Fd^−^ oxidation rate because Fd was kept oxidized more than 50% at the steady state of photosynthesis except for the condition where the electron acceptor side of PSI was extremely limited at low CO_2_ and O_2_ in mustard.

ECS was measured simultaneously with gas exchange using a Klas-100 spectrophotometer (Walz) [61]. The ECS signal was calculated from two pulse-modulated dual-wavelength difference signals using the following equation: (ΔI/I_521.4−507.6_ + ΔI/I_520.2−534.8_)/2. Red actinic light was temporarily turned off for 600 ms in a dark-interval relaxation kinetics analysis during the steady-state photosynthesis to determine ECS parameters [62]. The total rapid (<1 s) change in ECS signal upon rapidly switching off actinic light was defined as ECS_Total_. The parameter *v*_H_^+^ was estimated by a linear fitting for the initial phase of the mono-exponential decay of ECS_Total_ in the transition from light to dark, giving the rate constant of the decay *g*_H_^+^ by the calculation. The ΔpH component of ECS_Total_ was termed as ECS_ΔpH_ and was estimated as the ECS fraction recovered in 30 s after turning red actinic light off at the steady state of photosynthesis [31], which are used in Figure 5A,C. The amplitude of ECS was normalized by ECS_SSF_, the ECS change induced by a 5 µs-short saturation flash [35].

All in vivo spectroscopic measurements were based on the assumption that the absorption coefficient and the amplification factor [63] of each targeted molecule are not different between mesophyll and bundle sheath cells. All statistical analyses were performed using Origin 2017 (Lightstone, Tokyo, Japan).

## Figures and Tables

**Figure 7 ijms-22-04894-f007:**
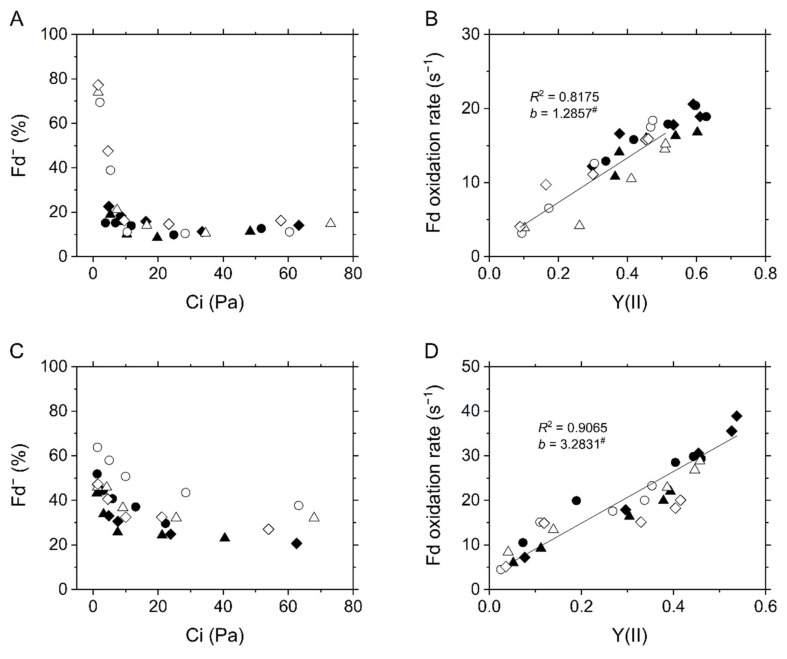
In vivo measurement for the redox state of ferredoxin (Fd) in the C_3_ plant mustard (**A**,**B**) and the C_4_ plant maize (**C**,**D**). (**A**,**C**) The Fd reduction during the steady-state photosynthesis at various intercellular CO_2_ partial pressures (Ci). (**B**,**D**) Relationship of Fd^−^ oxidation rate with effective quantum yield of PSII, Y(II), at various Ci. Experiments were conducted independently three times as shown in different symbols (biological replicates) at 21 kPa (closed symbols) and 1 kPa O_2_ (open symbols). Solid lines represent the estimated linear regressions of the data at 1 kPa (**B**) and 21 kPa O_2_ (**D**), respectively (*R*^2^, coefficient of determination). The *y*-intercepts (*b*) were tested based on the null hypothesis: ^#^ *p* > 0.05.

**Figure 8 ijms-22-04894-f008:**
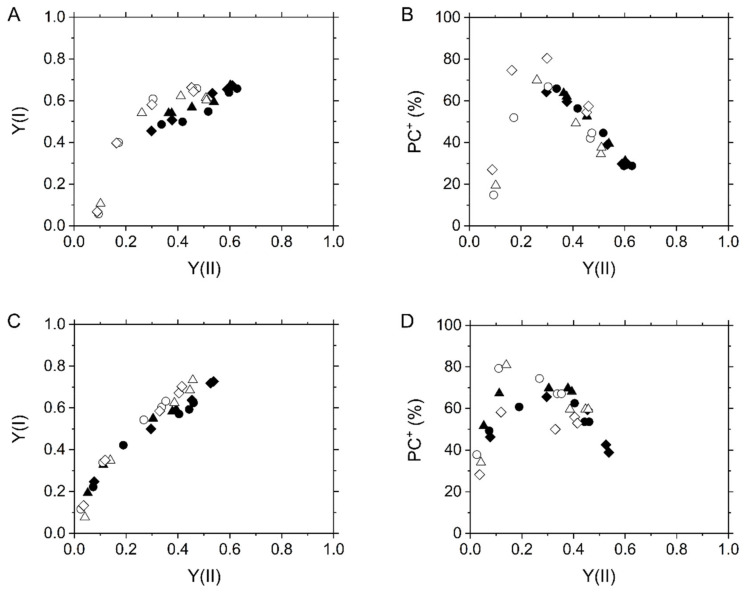
Relationships of effective quantum yield of PSI, Y(I) (**A**,**C**), and plastocyanin (PC) oxidation (**B**,**D**) with effective quantum yield of PSII, Y(II), at various intercellular CO_2_ partial pressures in the C_3_ plant mustard (**A**,**B**) and the C_4_ plant maize (**C**,**D**). Experiments were conducted independently three times as shown in different symbols (biological replicates) at 21 kPa (closed symbols) and 1 kPa O_2_ (open symbols).

## Data Availability

Not applicable.

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
