# Peer review of "Photosynthetic Linear Electron Flow Drives CO2 Assimilation in Maize Leaves"

_ijms, 2021, doi:10.3390/ijms22094894_

Round 1
Reviewer 1 Report
Review of manuscript no. 1174299 titled
“Photosynthetic linear electron flow drives CO2 assimilation in maize leaves”
by G. Shimakawa and C. Miyake
submitted for publication to International Journal of Molecular Sciences
The manuscript is interesting and suitable for publication in the journal. However, since I have some comments listed below, I suggest a revision of the manuscript.
General comments
Even if role of photorespiration is important for the topic explored in the manuscript, oxygenase activity of Rubisco was not measured.
The manuscript presents only one side view on the generation of reactive oxygen species (ROS) and damage of only PSI by ROS even if generation of ROS in PSI was not measured and oxygenation of PSI amino acids was not measured too. On the other hand, it is well-documented in the literature that ROS are generated in PSII as well as that PSII amino acids are oxidized by ROS (mentioned in one of my specific comments.
Specific comments:
Line 26, 74, … – The term „proton gradient“ is not correct, even if it is usually used terminology. A gradient is defined as a continual change in space. It is a difference of proton concentration across the thylakoid membrane, not proton gradient.
Line 58-62 – The term „instead of“ in the sentence gives the sentence a strange meaning. Thus, please, rephrase the sentence.
Line 67-70 – The sentence is not clear. Please, rephrase it; the sentence should contain more detail information why presence of P700+ is so important (which reactions are involved or which reactions are not happening when P700+ is present). Moreover, ROS can damage also PSII (in fact anything), see, e.g. Kale et al.(PNAS 114(11), 2988-2993, 2017) and Kumar et al. (PNAS 118(4), e2019246118, 2021). Consequently, if you study damage of PSI only in your manuscript, it is OK, but you should mention that also PSII and other proteins and also lipids can be damaged/oxidized by ROS.
L74 – To consider oxygenation of RuBP by Rubisco as reaction on the „acceptor side“ of PSI is strange to me since by the PSI acceptor side is usually ment Fx, Fa, Fb and Fd. Rubisco as a part of Calvin-Benson-Bassham cycle is not the acceptor side of PSI.
L78-80 – It is strange to me – to inhibit electron transport to P700+ by delta pH, protons must first accumulate in lumen. But transport of protons to lumen is always coupled with transport of electrons. Thus, accumulation of protons in lumen is always accompanied by transport of electrons to P700+ and thus, no P700+ will be accumulated. Or where is the point of the hypothesis?
L85,86 – Change „keeping …. meeting“ to „keep …. meet“.
L100 – Change “like” to “same as”
Fig 2A – There is a point of Y(II) ≈ 0.1 for Ci≈3 Pa for 21 kPa O2 in Fig 1B and this point is missing in Fig. 2A.
L135 – There should be “Fig. S1A” instead of “Fig. S1B”
L153 – You should mention somewhere here that ECStotal is the measured ECS signal normalized to the ECS signal caused by single-turnover flash.
L168 – If you describe dependence of vH+ on CO2 assimilation rate for the case of mustard as almost linear (Fig. 4C), than the dependence is also linear for the case of maize (Fig. 4F).
L174 – Change „field“ to „potential“ – electric field and electric potential are different variables.
L177 – It is not „formation of deltapH“ but „formation of deltapH fraction of pmf“.
L192 – The two-exponential fit of the decay shown in the Fig. S3 is poor.
L194-196 – Why you do not consider also the fact that reduced Fd is oxidized by O2 at high O2 pressures and low Ci? High O2 causes not only increased photorespiration but also increased Mehler reaction.
L237 – The term “connectivity” is usually related to energetic connectivity in the literature, which probably not your case. Thus, specify which connectivity.
L263 – What do you mean by “Oxidation of P700”? Probably “keeping P700 oxidized”. If this is the case, then it is strange why, according to you theory, a plant tries to have always P700+ state when it is the P700 state which is necessary to be excited and consequently to release electron to subsequent PSI electron acceptors to run photosynthesis. If P700 is always oxidized, there is no time to run regular electron transport through PSI. If there is no regular charge separation in PSI, there is no regular photosynthetic function of whole system. Thus, you should pay more attention in you manuscript to explain in more detail a necessity of accumulation of P700+ for photosynthetic function.
L331-333 – Cannot involvement of the Q-cycle of cyt b6/f be the case?
L435 – Measurement of the deltapH faction of pmf is not sufficiently described, thus, the description should be improved.
Author Response
Even if role of photorespiration is important for the topic explored in the manuscript, oxygenase activity of Rubisco was not measured.
The manuscript presents only one side view on the generation of reactive oxygen species (ROS) and damage of only PSI by ROS even if generation of ROS in PSI was not measured and oxygenation of PSI amino acids was not measured too. On the other hand, it is well-documented in the literature that ROS are generated in PSII as well as that PSII amino acids are oxidized by ROS (mentioned in one of my specific comments.
>> Thank you for your comments. We totally revised the manuscript following your specific advices.
Specific comments:
Line 26, 74, … – The term „proton gradient“ is not correct, even if it is usually used terminology. A gradient is defined as a continual change in space. It is a difference of proton concentration across the thylakoid membrane, not proton gradient.
>> We changed “proton gradient” to “difference of proton concentration” throughout the manuscript.
Line 58-62 – The term „instead of“ in the sentence gives the sentence a strange meaning. Thus, please, rephrase the sentence.
>> We revised the sentence (line 59-60).
Line 67-70 – The sentence is not clear. Please, rephrase it; the sentence should contain more detail information why presence of P700+ is so important (which reactions are involved or which reactions are not happening when P700+ is present). Moreover, ROS can damage also PSII (in fact anything), see, e.g. Kale et al.(PNAS 114(11), 2988-2993, 2017) and Kumar et al. (PNAS 118(4), e2019246118, 2021). Consequently, if you study damage of PSI only in your manuscript, it is OK, but you should mention that also PSII and other proteins and also lipids can be damaged/oxidized by ROS.
>> In the indicated paragraph, we discuss PSI photoinhibition only. However, we agree that we should note that PSII is damaged by ROS (more than PSI). In the revised manuscript, we rephrased the sentences (line 69-72).
L74 – To consider oxygenation of RuBP by Rubisco as reaction on the „acceptor side“ of PSI is strange to me since by the PSI acceptor side is usually ment Fx, Fa, Fb and Fd. Rubisco as a part of Calvin-Benson-Bassham cycle is not the acceptor side of PSI.
>> We withdrew the definition “the acceptor-side mechanism” to photorespiration and just mentioned it as a regulatory mechanism related to P700 oxidation in the revised manuscript.
L78-80 – It is strange to me – to inhibit electron transport to P700+ by delta pH, protons must first accumulate in lumen. But transport of protons to lumen is always coupled with transport of electrons. Thus, accumulation of protons in lumen is always accompanied by transport of electrons to P700+ and thus, no P700+ will be accumulated. Or where is the point of the hypothesis?
>> As you mentioned, the pumping proton to the luminal side of the thylakoid membrane is coupled with photosynthetic linear electron transport. Also, the proton efflux to the stroma side is mainly mediated by ATP synthase. The point is that the proton concentration in the lumen depends on the input/output of the fluxes (more strictly, which the limitation step of the proton flux is). There is much unknown in the regulation of luminal proton concentration. One hypothesis is that cyclic electron flow supports an additional pumping of proton into the lumen, resulting in the increase in the proton concentration. The other hypothesis is the regulatory mechanism that limits proton efflux via ATP synthase, also resulting in the increase of the luminal proton concentration. The physiological results in our present work clearly support the latter hypothesis.
L85,86 – Change „keeping …. meeting“ to „keep …. meet“.
>> We revised the sentence (line 91).
L100 – Change “like” to “same as”
>> We revised the sentence (line 105).
Fig 2A – There is a point of Y(II) ≈ 0.1 for Ci≈3 Pa for 21 kPa O2 in Fig 1B and this point is missing in Fig. 2A.
>> We could not find the point of Y(II) ≈ 0.1 for Ci≈3 Pa for 21 kPa O2 in Fig 1B.
L135 – There should be “Fig. S1A” instead of “Fig. S1B”
>> We revised the sentence (line 139).
L153 – You should mention somewhere here that ECStotal is the measured ECS signal normalized to the ECS signal caused by single-turnover flash.
>> We added the explanation (line 155-156).
L168 – If you describe dependence of vH+ on CO2 assimilation rate for the case of mustard as almost linear (Fig. 4C), then the dependence is also linear for the case of maize (Fig. 4F).
>> We revised the sentences in reference to your comment (line 169, 174-175).
L174 – Change „field“ to „potential“ – electric field and electric potential are different variables.
>> We revised the sentence (line 180).
L177 – It is not „formation of deltapH“ but „formation of deltapH fraction of pmf“.
>> We revised the sentence (line 183-184).
L192 – The two-exponential fit of the decay shown in the Fig. S3 is poor.
>> We removed the figure.
L194-196 – Why you do not consider also the fact that reduced Fd is oxidized by O2 at high O2 pressures and low Ci? High O2 causes not only increased photorespiration but also increased Mehler reaction.
>> We agree that O2 is reduced to superoxide radical by Mehler reaction in the acceptor side of PSI, which is probably enhanced with the electron acceptor-side limitation of PSI. However, on the view of the electron flux capacity, photorespiration is obviously dominant as the electron sink (for example, Driever and Baker 2011, Sejima et al. 2016). Therefore, we suggest that photorespiration mainly functions to relieve the acceptor-side limitation at the steady state photosynthesis under CO2 limitation. In the other word, the electron flux via Mehler reaction is very small, compared with photosynthetic linear electron transport, but the generated ROS have the significant impact on the photo-oxidative stress. We added this explanation to the revised manuscript (line 202-206).
L237 – The term “connectivity” is usually related to energetic connectivity in the literature, which probably not your case. Thus, specify which connectivity.
>> We revised the sentences (line 246, 261)
L263 – What do you mean by “Oxidation of P700”? Probably “keeping P700 oxidized”. If this is the case, then it is strange why, according to you theory, a plant tries to have always P700+ state when it is the P700 state which is necessary to be excited and consequently to release electron to subsequent PSI electron acceptors to run photosynthesis. If P700 is always oxidized, there is no time to run regular electron transport through PSI. If there is no regular charge separation in PSI, there is no regular photosynthetic function of whole system. Thus, you should pay more attention in you manuscript to explain in more detail a necessity of accumulation of P700+ for photosynthetic function.
>> The term “P700 oxidation” means “keeping P700 oxidized”. We explained the details of the term in the Introduction section (line 72-75).
L331-333 – Cannot involvement of the Q-cycle of cyt b6/f be the case?
>> If the additional H+-pumping mechanism that we hypothesized is related to the Q-cycle, it is very interesting to discuss. However, unfortunately, the molecular mechanism is unclear now, and we would not like to propose any novel mechanisms in that section.
L435 – Measurement of the deltapH faction of pmf is not sufficiently described, thus, the description should be improved.
>> We added the explanation in the revised manuscript (line 441-442).
Reviewer 2 Report
The paper by Shimakawa and Miyake presents an interesting view on the role and importance of different regulatory mechanisms in the photosynthetic electron transport system of a C4 plant as compared with a C3 plant. The paper supports the presented ideas strongly with a broad range of experimental results. I think the following changes could improve the reception of the paper.
Linguistic requests:
Since I find the results presented in the paper very interesting it is especially important that the authors make some additional efforts to improve the clarity and readability of the paper.
I suggest a careful linguistic revision of the entire text, with special attention to compound sentences with multiple clauses. In these sentences it is often difficult to follow the logical connection of the different clauses. Please see lines 58-62, 94-97, 99-102, 215-217, 251-253, 259-263, 277-279, or even in the abstract e.g. lines 22-24, 27-31.
In line 117 does the statement refer to the 1kPa or the 21kPa O2 curves or both of the curves in 1B and C?
Line 135: Please check if Fig. S1A is to be used instead of S1B.
Line 178: Please clarify in the sentence if very low Ci refers to the case of only maize or both plants.
Scientific requests:
Lines 197-198: The statement “regardless of O2” seems to be supported by Fig. 7B, but not by Fig. 6B. Please clarify this.
In Fig. 8. A. few data points at low Y(II) do not satisfy the statement of line 211 “resulting in the extra Y(I)”. Authors should comment on this.
The statement in Lines 217-218 “Interestingly, the extra Y(I) to Y(II) was coincided with the oxidation of PC (Fig. 8).” is an important element of the conclusion so should be demonstrated by and independent graph where (Y(I)-Y(II)) is plotted (e.g. as a function of PC+ or along PC+ as a function of Y(II) or some other clear way preferred by the authors).
Minor comments:
Is “the” required in line 235?
Author Response
The paper by Shimakawa and Miyake presents an interesting view on the role and importance of different regulatory mechanisms in the photosynthetic electron transport system of a C4 plant as compared with a C3 plant. The paper supports the presented ideas strongly with a broad range of experimental results. I think the following changes could improve the reception of the paper.
>> Thank you for your comments. We totally revised the manuscript in reference to your advices.
Linguistic requests:
Since I find the results presented in the paper very interesting it is especially important that the authors make some additional efforts to improve the clarity and readability of the paper.
I suggest a careful linguistic revision of the entire text, with special attention to compound sentences with multiple clauses. In these sentences it is often difficult to follow the logical connection of the different clauses. Please see lines 58-62, 94-97, 99-102, 215-217, 251-253, 259-263, 277-279, or even in the abstract e.g. lines 22-24, 27-31.
>> We revised these sentences (line 23-24, 28-29, 59-60, 100-102, 104, 224-225, 261, 268-272, 287-288).
In line 117 does the statement refer to the 1kPa or the 21kPa O2 curves or both of the curves in 1B and C?
>> We added “at 21 kPa O2” (line 121).
Line 135: Please check if Fig. S1A is to be used instead of S1B.
>> We corrected the sentence (line 139).
Line 178: Please clarify in the sentence if very low Ci refers to the case of only maize or both plants.
>> We clarified the sentence (line 184).
Scientific requests:
Lines 197-198: The statement “regardless of O2” seems to be supported by Fig. 7B, but not by Fig. 6B. Please clarify this.
>> We clarified the sentence (line 207-208).
In Fig. 8. A. few data points at low Y(II) do not satisfy the statement of line 211 “resulting in the extra Y(I)”. Authors should comment on this.
>> As you suggest, a few points at low Y(II) did not show the extra Y(I) because at these low Y(II) PC is kept reduced. In the revised manuscript, we produced additional figures to clarify this in reference to your advice.
The statement in Lines 217-218 “Interestingly, the extra Y(I) to Y(II) was coincided with the oxidation of PC (Fig. 8).” is an important element of the conclusion so should be demonstrated by and independent graph where (Y(I)-Y(II)) is plotted (e.g. as a function of PC+ or along PC+ as a function of Y(II) or some other clear way preferred by the authors).
>> We newly plotted Supplemental Fig. S3 in the revised manuscript.
Minor comments:
Is “the” required in line 235?
>> We removed “the” in these sentences (line 242, 245).